# Comparison of the Chemical Composition of Selected Varieties of Elderberry with Wild Growing Elderberry

**DOI:** 10.3390/molecules27165050

**Published:** 2022-08-09

**Authors:** Agnieszka Nawirska-Olszańska, Maciej Oziembłowski, Pavla Brandova, Marta Czaplicka

**Affiliations:** 1Department of Fruit, Vegetable and Plant Nutraceutical Technology, Wroclaw University of Environmental and Life Sciences, 50-375 Wrocław, Poland; 2Department of Functional Food Products Development, Wroclaw University of Environmental and Life Sciences, 50-375 Wrocław, Poland; 3Research and Breeding Institute of Pomology, 508 01 Holovousy, Czech Republic; 4Department of Horticulture, Wroclaw University of Environmental and Life Sciences, 50-375 Wrocław, Poland

**Keywords:** *Sambucus nigra*, carotenoids, triterpenoids, sugar, organic acids

## Abstract

Elderberries of wild-growing shrubs are most often used; however, various cultivated varieties of this shrub appear more and more often. The aim of this research was to compare the fruit composition of specific varieties with those grown wild in urban and ecologically clean conditions. Six varieties of elderberry grown on one experimental farm and two wild-growing samples from the city center and the landscape park were assessed. The content of vitamin C, antioxidant activity, sugar and organic acid content, triterpenes and carotenoids was marked in the tested fruits. The analyses show that there were significant differences in the content of the tested ingredients between the varieties tested, while the place of cultivation was of less importance. Apart from organic acids and triterpenes, fruits from wild-growing shrubs were more abundant in other compounds determined. The white variety of ‘Albida’ turned out to be the poorest in bioactive compounds.

## 1. Introduction

*Sambucus nigra* L., which belongs to the Caprifoliaceae family, is a shrub found growing on roadsides and wasteland across Europe as well as in southwestern Asia and North America. The wide range of prevalence of elderberries in Europe results from the way in which seeds of this species (ornitochoria) are spread and from high resistance to environmental conditions [1]. Although it has been known in folk medicine for centuries, it is often considered to be a less-interesting plant, not worthy of attention. This is too bad because elderberry can be used in gardens as an ornamental plant and in plantations as a source of flowers, leaves, bark and fruits with valuable economic and healthy properties [2]. Elderberry is a bush with edible fruit that is possible to cultivate in different types of soil. They provide good yields in weak soil, with not much water. Regarding *Sambucus nigra* L., it is often possible to grow flowers and fruit in urban areas. The fruit received from ornamental forms of bushes is also edible. It is one of the most popular plants grown in urban areas, with high antioxidant activity [3]. The possibility of obtaining edible fruits with a high content of valuable compounds from organized areas makes this plant interesting for further research

In Europe, there are three species of the genus Sambucus: red elderberry (*S. racemosa*), a shrub with poisonous coral red fruits; Sambucus ebulus (*S. ebulus*), a perennial with no woody shoots and inedible black fruit; black elderberry (*S. nigra*), a shrub with edible black fruits.

Elderberry has been used for many years in medicine to treat many diseases and is also used in food supplements supporting immuno-resistance. For medical use, flowers, fruits, leaves and bark are used.

In recent times an interest in elderberry has been growing. Almost all parts of the bush (bark, leaves, flowers and fruits) can be used in manufacturing, and most have medicinal properties. It is treated as a source of substances with antioxidant and anti-cancer properties [4]. Fruit puree is often used as a natural dye in the food industry, while the juice is used for the preparation of medicinal syrups.

Elderberry fruits have diaphoretic, diuretic and laxative properties (which makes it easier to remove toxins from the body). In addition, they have analgesic activity and are used in treating inflammation of the trigeminal nerve, sciatica and migraines, as well as cancer [4].

The fruits of elderberry have vitamins A, B and C. The vitamin C content varies from 6 to 35 mg/100 g depending on the variety [5]. In the fruits of different varieties, hybrids and wild elderberries, 5 organic acids, 19 anthocyanins, 11 flavanols and 9 polyphenolic acids were identified; their quality and quantity were varied depending on the variety and place of harvest [6,7,8]. In the fruits, three triterpenes were also identified [9]; the amount depended both on the cultivar and harvest date. The ash content of the fruit was approx. 0.99%, containing minerals such as K, Ca, Fe, Mg, P, Na, Zn, Cu, Mn, Se, Cr, Ni and Cd [10,11,12].

The nutrient composition of raw elderberries per 100 g edible portion was reported as: water 79.80 g, energy 73 kcal (305 kJ), protein 0.66 g, total lipid 0.50 g, ash 0.64 g, carbohydrate 18.40 g, total dietary fiber 7.0 g, Ca 38 mg, Fe 1.60 mg, Mg 5 mg, P 39 mg, K 280 mg, Na 6 mg, Zn 0.11 mg, Cu 0.061 mg, Se 0.6 mg, vitamin C 36 mg, thiamin 0.070 mg, riboflavin 0.060 mg, niacin 0.5 mg, pantothenic acid 0.140 mg, vitamin B-6 0.230 mg, total folate 6 mg, vitamin A 30 mg RAE, vitamin A 600 IU, total saturated fatty acids 0.023 g, 16:0 (palmitic acid) 0.018 g, 18:0 (stearic acid) 0.005 g, total monounsaturated fatty acids 0.080 g, 18:1 undifferentiated 0.080 g, total polyunsaturated fatty acids 0.247 g, 18:2 undifferentiated 0.162 g, 18:3 undifferentiated 0.085 g, tryptophan 0.013 g, threonine 0.027 g, isoleucine 0.027 g, leucine 0.060 g, lysine 0.026 g, mehtionine 0.014 g, cysteine 0.015 g, phenylalanine 0.040 g, tyrosine 0.051 g, valine 0.033 g, arginine 0.047 g, histidine 0.015 g, alanine 0.030 g, aspartic acid 0.058 g, glutamic acid 0.096 g, glycine 0.036 g, proline 0.025 g and serine 0.032 g [12]. Wild European elderberry plants in Turkey were found to be very rich in terms of health components, with a high protein content of 2.68–2.91%, a high total antioxidant capacity of 6.37 mmol/100 g FW, total phenolic content of 6432 mg GAE/100 g FW and total anthocyanin content of 283 mg cyaniding-3-glucoside/100 g FW [5,12].

Fruits are characterized by a high content of anthocyanins, which give them their characteristic dark-purple color [11,13]. Anthocyanins have anti-inflammatory, antiallergic properties, are soothing, improve blood circulation in the capillaries, strengthen the walls of blood vessels, reduce oxidative damage [14,15] and are used as natural pigments in the food industry [16].

Anthocyanins, as well as other flavonoids (e.g., quercetins), exhibit antioxidant, anticarcinogenic, immune-stimulating, antibacterial, antiallergic and antiviral properties; therefore, their consumption may contribute to the prevention of several degenerative diseases such as cardiovascular disease, cancer, inflammatory disease and diabetes [11]. They have gained interest as functional compounds in food colorants and as potent agents against oxidative stress, reducing oxidative damage to the human body [11].

In the literature on the subject, there are many works devoted to the elderberry fruit. Some describe selected varieties thereof. However, these works are devoted mainly to the content of polyphenols, and in particular anthocyanins, in the tested fruits or products. However, there are not many studies comparing the cultivated varieties with the wild-growing ones. As it is known, the content of bioactive compounds in a plant depends on many factors. In addition to the variety of a given species, the growing sites play a very important role. The aim of the research was to compare the chemical composition and the content of bioactive compounds of six varieties of the elderberry grown in the same soil and climatic conditions with the varieties growing wild in the area of the Landscape Park and in the industrialized region. Plants were grown in a natural, bushy form. All the bushes were fully fruiting. Their age is 5–10 years. The plants were grown on sandy, light soils, surrounded by other fruit crops.

## 2. Results and Discussion

### 2.1. Basic Determinations and Antioxidant Activity

The dry matter content in the tested elderberry cultivars ranged from 17.71% to 22.32%, as shown in Table 1. In the studies of Vulić, Vračar and Šumić [17], the dry matter content was on a similar level at 20.22%. In the studies of Diviš, Pořizka, Vespalcová, Matějiček and Kaplan [18], employing the ‘Albida’ and ‘Bohatka’ varieties, they obtained a lower dry matter content. The difference was due to different cultivation years. The collection site had no major effect on the dry matter content; it was more of a variety-dependent trait. The results of the ash content in the tested samples were similar. A greater influence of the variety on its content was noticed than the place of its collection. Its content ranged from 0.88 to 1.28%. The acidity in the tested samples ranged from 0.52 g/100 g FW to 0.93 g/100 g FW. The ash content, as well as the acidity determined by Vulić, Vračar and Šumić [17], were at a similar level, while the pectin content in these tested cultivars was significantly higher. In the presented studies, the pectin content ranged from 0.83% to 1.54%, and in the studies, by Vulić, Vračar and Šumić [17], it was only 0.1593%. The highest amount of pectins was determined in both variants of wild-growing elderberry shrubs and the lowest in the white variety of ‘Albida’ (0.83%).

The content of vitamin C, determined as the content of ascorbic acid, was the highest in the wild-growing varieties, 39.67 mg/100 g FW and 44.26 mg/100 g FW in cultivars, which was significantly less, from 8.56 mg/100 g FW to 23.28 mg/100 g FW. A smaller amount of vitamin C was determined in the fruits from the Landscape Park, although both collection sites formed a homogeneous group. In the studies by Vulić, J, Vračar, L. and Šumić [17], vitamin C was determined at the level of 34.10 mg/100 g. The vitamin C content largely depends on the degree of maturity. Nevertheless, the three-year averages of the studies show that wild crops have higher vitamin C content than the specific regular crops tested.

The antioxidant activity of different variants of elderberry was measured by the radical cation decolorization assay using the ABTS and ferric reducing capacity and the FRAP method (Table 1). The highest ABTS and FRAP values were 3359.04 and 7272.07 µmol/100 g FW, respectively. The highest antioxidant activity was determined in the Sambo variety, while the lowest activity was demonstrated by the white elderberry fruit of the ‘Albida’ variety. Fruits from wild-growing shrubs had similar antioxidant activity and were at the end of the results for the cultivated varieties. There were no statistical differences between the two selected fruit picking sites. Less activity was determined in the cultivars of ‘Samdal’ and ‘Albida’.

### 2.2. Sugars and Organic Acid

Six varieties were grown in one place, two other collection sites of elderberry were examined in this study and the concentrations of individual sugars were assessed. The most abundant sugars in black elderberry fruit were fructose and glucose; sorbitol was detected only in small amounts in the white variety of elderberry, ‘Albida’. The individual and total sugar content in elderberry fruit was highest in ‘Wildly growing P’ and ‘Wildly growing C’, with average values of 5.85 g/100 g and 5.26 g/100 g fresh weight (FW) (Table 2). The lowest sugar content was determined in the ‘Bohatka’ variety, only 1.11 g/100 g FW. The collection site, the Landscape Park or the city did not affect the sugar content. The wild fruits were richer in these compounds than those from regular cultivation. Taking into account only these varieties, the highest amount of sugars was determined in the variety of ‘Albida’, in which only small amounts of sorbitol were determined. In this variety, the highest amount of fructose among the tested fruits was also determined at 2.84 g/100 g FW. In these studies, the sugar content was significantly lower than in the studies carried out by Veberic, Jakopic, Stampar and Schmitzer [6]. In the ‘Haschberg’ variety, they determined that the sugar content was twice as high (6.85 g/100 g FW). Additionally, sucrose was marked in their research. In these studies, the obtained results cannot be due to the time of harvest and the degree of maturity of the elderberry fruit, as the fruits in individual elderberry clusters ripen unevenly. It sometimes occurs that in individual clusters, a greater amount of fruit is even less ripe; however, this is only determined after harvesting.

Four organic acids were identified in the elderberry fruit: citric acid, malic acid, shikimic acid and fumaric acid (Table 2). Citric acid was the most abundant organic acid in all tested fruits, followed by malic acid and smaller concentrations of shikimic and fumaric acid. The content level of citric acid in the fruit of elderberry ranged from 502.72 mg/100 g FW in ‘Bohatka’ to 265.65 g/100 g FW in the variety ‘Albida’.

The ‘Bohatka’, ‘Sambo’ and ‘Haschberg’ varieties (658.94, 618.46 and 591.67 mg/100 g FW, respectively) were characterized by the highest sum of acids, and they constituted one homogeneous group. Similar results in their research were obtained by Veberic, Jakopic, Stampar and Schmitzer [6]. The wild-growing variety harvested in two sites was characterized by a low acid content comparable to the ‘Samdal’ variety but slightly higher than the ‘Albida’ variety, which contained the lowest amount of organic acids and sugars.

### 2.3. Triterpenes

Table 3 show the levels of betulinic, oleanolic and ursolic acid of the investigated elderberries. Ursolic acid was a more abundant triterpene compound than oleanolic and betulinic acid in all investigated variants of elderberry berries. The greatest amount of total triterpenes was determined in the white elderberry variety of ‘Albida’ (1574.13 µg/g FW), while the least was determined in the variety of ‘Samdal’, quite popular among growers (538.28 µg/g FW). The place of the collection did not matter; ‘Weihenstephan’ contained almost the same amount of total triterpenes as the wild-shrub fruit harvested in the city center. In the fruit harvested in the Landscape Park, their content was slightly lower (1049.61 µg/g FW). However, the content of individual acids in these cases was different (Table 3).

The presence of triterpenes in the elderberry fruits was confirmed by the research of Salvador, Rochaa and Silvestre [9] in three varieties from Portugal. The researchers identified three triterpenes, β-Amyrin, Ursolic acid and Oleanolic acid, and the fourth was not identified by them. Similarly to the present study, Ursolic acid was the most numerous and the least unidentified; it should be presumed that the most identified was Betulinic acid. In this study, Amyrin, β-Amyrin, was not determined, which may be due to the small amounts found in elderberries.

In the research of Gleńsk et al. [19], the effect of elderberry triterpenes on the inhibition of cancer cell growth was demonstrated. These studies confirmed that ursolic acid is the dominant triterpenoid in the elderberry fruit. Three triterpenoids were also determined in Saskatoon berries; the most prevalent was Oleanolic acid and the least was Ursolic acid [20]. It follows that a different triterpene is dominant in the elderberry fruit, and another is dominant in Saskatoon berries. 

### 2.4. Carotenoids

In the available literature on the subject, there is no data on the content of carotenoids in the selected varieties of elderberries. Due to their properties, carotenoids are one of the most valuable bioactive compounds found in fruits and vegetables.

Four carotenoids were identified in the varieties tested: all-trans-lutein, β-cryptoxanthine, α-carotene and 15-cis-β-carotene (Table 4). The dominant carotenoid was β-cryptoxanthine, the highest amounts of which were found in both variants of wild shrub fruit (wild-growing P—841.61 mg/kg FW and wild-growing C—789.58 mg/kg FW). At the same time, in both variants of wild-growing fruit, the overall content of carotenoids was the highest. As in the case of β-cryptoxanthines and other carotenoids, they constituted a separate homogeneous group. In the ‘Albida’ variety, α-carotene and 15-cis-β-carotene were not determined, and it was the variety with the lowest total carotenoid content. However, in the ‘Bohatka’ variety, an exceptionally large number of 15-cis-β-carotene was determined at 29.55 mg/kg FW, with a mean of 5.86 mg/kg FW for the remaining samples. The elderberry fruits were, therefore, quite rich in carotenoids. In the fruits of the Saskatoon elderberry, four carotenoids (zeaxanthin, all-trans-lutein, 13-cis*yew*-lutein, β-carotene) with seven Saskatoon genotypes in all subjects were determined. The highest numbers were determined for β-carotene and the lowest for 13-*cis*-lutein [20].

The analysis of Figure 1 shows the existence of two compact clusters of points representing the analyzed varieties. The first group consists of the cultivars Samdal, Weihenstephan, Haschberg, Sambo and Bohatka, while the second group is wild-growing C and P. Apart from these two groups, there was also the Albida cultivar, which differed most from the other two groups in terms of the chemical composition of its fruits.

The group of acids (including their sum) had a similar high and positive effect on the value of the first component (PCA1), and the strongest negative impact on this value was, in turn, due to fructose, the sum of sugars and sorbitol. The sorbitol content had a positive effect on the value of the second component (PCA2), while the ash content and the sum of triterpenoids had a similar effect to a lesser extent. The following compounds had a strong negative effect on the second component: vitamin C, beta-cryptoxanthin, trans-lutein and the sum of carotenoids. Correlations between selected variables can also be read from the figure (Figure 2). For example, the variables related to acids, to which the fructose content was high and negatively correlated, were characterized by a high positive correlation. On the other hand, the values of carotenoids were very poorly correlated with the values of FRAP and ABTS.

## 3. Materials and Methods

### 3.1. Reagents and Standard

Acetonitrile, formic acid, methanol, ABTS, Trolox Equivalents (6-hydroxy-2,5,7,8-tetramethylchroman-2-carboxylic acid), TPTZ (2,4,6-tris (2-pyridyl)-s-triazine), acetic acid, phloroglucinol, caffeic acid, betulinic, oleanolic and ursolic acid, fructose, glucose and sorbitol, malic, citric, shikimic, and fumaric acid, 15-cis-β-carotene, α-carotene, all-trans-Lutein and β-cryptoxanthin were purchased from Sigma-Aldrich (Steinheim, Germany).

### 3.2. Plant Materials

The cultivars studied come from the Research and Breeding Institute in Holovousy in the Czech Republic, and concerning the wild-growing plants, one of them comes from the Śnieżnik Landscape Park (the wild-growing P) and the other from the urbanized region, the center of Wrocław (the wild-growing C) in Poland. The fruit was harvested when it was ripe for consumption in 2017–2019. ‘Albida’, ‘Bohatka’ and ‘Sambo’ are Slovak varieties, ‘Haschberg’ is an Austrian variety, ‘Samdal’ comes from Denmark and ‘Weihenstephan’ is a German variety.

All plants grew naturally, accompanying other fruit crops. No additional fertilization or plant protection was employed. The fruits were harvested in the second week of September after all the varieties were fully ripe. Maturity was assessed on the basis of easy fruit detachment from the stalk. Twelve plants were sampled for each cultivar, and the tests were carried out with four replications per combination. After harvesting, the fruit was frozen, and then the homogenization of the plant material was commenced for the purpose of conducting tests. Whole fruits were analyzed without petals.

The course of the weather in the research years did not differ significantly from the long-term averages. The average temperature in July in all surveyed sites was between 18 and 19.5 °C. There were no periods of drought or excessive rainfall.

The presented results are average data from three years of sampling (2019–2021). The same analyses were performed each year and then averaged in order to present the results of studies representative of three fruiting seasons.

### 3.3. Dry-Matter, Ash Content, Titratable Acidity, Pectin and Vitamin C

Assessment of the dry-matter content of the elderberry fruit was performed by means of a gravimetric method using the standard AOAC method [21]. The fresh elderberry samples were precisely weighed (1.5 g) and dried at a temperature of 70 °C under 3 kPa vacuum pressure until a constant weight was obtained. Measurements were performed in triplicate and expressed as a %.

Fruit ash content was determined using the Association of Official Analytical Chemists [21] 930.09 method. Measurements were performed in triplicate and expressed as a %.

The titratable acidity of the samples was determined using a pH meter (IQ’s Scientific Instruments), according to Polish Norm [22]. The chopped apples were transferred to a volumetric flask (100 mL) and filled with water. Prepared samples were boiled and filtered after cooling down; about 10 mL of obtained filtrate was titrated with NaOH (0.1 mol/dm^3^) up to pH 8.1. The measurements were performed in triplicate and expressed in g of malic acid per100 g of FW.

Pectin, determined by the method of Morris, weighed approx. 10 g of the sample; 50 mL of distilled water were added, then the mixture was boiled for 30 min and filtered afterwards. The obtained filtrate was transferred to a 50 mL volumetric flask and filled with distilled water up to the marking. In total, 10 mL of the solution was mixed with 50 mL of acetone, and the test was left for 1 h. It was then filtered through a dried and weighed filter and then dried at 75 °C for 4 h and weighed. Pectin content was calculated from the difference between the weight of the filter with the sediment after drying and the weight of the dried filter, considering the amount of pectin solution for determination and the sample weight. The result was provided in % [23].

Vitamin C was analyzed in accordance with Polish Norm [24]. The method consists of the oxidation of l-ascorbic acid to dehydroascorbic acid in an acid medium with a blue dye of 2,6-dichloroindophenol, followed by the reduction of the dye to the leuko (colorless) form, which takes on a red color at a pH of 4.2.

### 3.4. Antioxidant-Activity Analysis

The determination of the ABTS and FRAP content was performed in methanol extracts (80% *v*/*v*; material to extracting agent ratio was 1:5). ABTS and FRAP antioxidant assays were carried out as previously described by Benzie and Strain [25] and Re et al. [26], respectively, using a UV-2401 PC spectrophotometer (Shimadzu Corp., Kyoto, Japan). All assessments were performed in triplicate. The results were expressed in µmol Trolox Equivalents per 100 g FW.

### 3.5. Analysis of Sugars with HPLC-ELSD Method

The extract for sugar analysis was prepared as described by Kolniak-Ostek [27]. Chromatographic analysis was carried out with an L-7455 liquid chromatograph (Merck-Hitachi, Tokyo, Japan) with an evaporative light scattering detector (PL-ELS 1000; Polymer Laboratories Ltd., Church Stretton, UK) and an L-7100 quaternary pump (Merck-Hitachi, Tokyo, Japan), equipped with a D-7000 HSM Multisolvent Delivery System (Merck-Hitachi, Tokyo, Japan), an L-7200 autosampler (Merck-Hitachi, Tokyo, Japan) and a Prevail Carbohydrate ES HPLC Column-W (250 × 4.6 mm, 5 µm; Alltech Inc., Nicholasville, KY, USA). Calibration curves (R2 = 0.9999) were created for glucose, fructose and sorbitol. All data were obtained in triplicate. The results were expressed as grams per 100 g of fresh weight (FW).

### 3.6. Analysis of Organic Acids by HPLC Method

For the extraction of samples, the protocol described by Nawirska-Olszańska et al. [28] was followed. Organic acid analysis was carried out with a Dionex (Sunnyvale, CA, USA) liquid chromatographer equipped with an Ultimate 3000 LED detector, LPG-3400A pump, EWPS-3000SI autosampler, TCC-3000SD column thermostat, an Aminex HPH-87 H (300 mm × 7.8 mm) column with IG Cation H (30 mm × 4.6 mm) Bio-Red precolumn and Chromeleon v.6.8 computer software. Separation was conducted at 65 °C. The elution solvent was 0.001 N sulfuric acid. Samples (10 μL) were eluted isocratically at a flow rate of 0.6 mL min^−1^. The compounds were monitored at 210 nm. All data were carried out in triplicate. The results were expressed as mg per 100 g of fresh weight (FW).

### 3.7. Analysis of Triterpenoids by the UPLC-PDA-MS/MS Method

Sample extraction was performed as described by Kolniak-Ostek, [27]. The identification and quantification of ursolic, oleanolic and betulinic acids was performed using an ACQUITY Ultra Performance LC system with a binary solvent manager (Waters Corp., Milford, MA, USA), a UPLC BEH C18 column (1.7 µm, 2.1 mm × 150 mm, Waters Corp., Milford, MA, USA) and a Q-Tof Micro mass spectrometer (Waters, Manchester, UK) equipped with an ESI source, operating in negative mode. The elution solvent was methanol-acetonitrile (15:85, *v*/*v*), at a flow rate of 0.1 mL min^−1^. The *m*/*z* for betulinic acid was 455.3452, for oleanolic acid it was 455.3496 and ursolic acid it was 455.3365, and the retention times were 6.80, 7.50 and 7.85 min, respectively. The compounds were monitored at 210 nm. All data were obtained in triplicate. The results were expressed as micrograms per gram of fresh weight (FW).

### 3.8. Carotenoid Content UPLC−PDA−MS Analysis

For the extraction and determination of carotenoids, a protocol described earlier was followed [20]. The powder samples of fruits (0.25 g) containing 10% MgCO_3_ were continuously shaken with 5 mL of hexane:acetone:methanol (2:1:1, *v*/*v*/*v*) containing 1% BHT at 500 rpm (DOS-10L Digital Orbital Shaker, Elmi Ltd., Riga, Latvia) for 30 min in the dark. After the first extraction, the samples were centrifuged at 19,000 g for 10 min at 4 °C, and the supernatant was recovered. The samples were re-extracted and centrifuged in the same conditions. Supernatants were combined and evaporated to dryness. The pellet was re-extracted using 2 mL of 100% methanol, filtered through a hydrophilic PTFE 0.20 µm membrane (Millex Samplicity Filter, Merck) and used for analysis.

Compounds were separated with an ACQUITY UPLC BEH RP C18 column (1.7 µm, 2.1 mm × 100 mm, Waters Corp.) at 32 °C. The elution solvents were ACN:MeOH (7:3, *v*/*v*) (A) and 0.1% formic acid (B). Samples (10 µL) were eluted according to the linear gradient described previously [20]. Weak and strong needle solvents were ACN–MeOH (7:3, *v*/*v*) and 2-propanol, respectively.

The identification of carotenoids was carried out on the basis of fragmentation patterns and on the basis of PDA profiles. Where available, compounds were compared with authentic standards (their fragmentation pathways, retention times and PDA profiles). The runs were monitored at 450 and 427 nm. The PDA spectra were measured over the wavelength range of 200–800 nm in steps of 2 nm. Calibration curves were developed from all-trans-β-carotene and all-trans-lutein. All incubations were carried out in triplicate. The results were expressed as milligrams per kilogram of fresh weight (FW).

### 3.9. Statistical Analysis

The obtained data were subjected to statistical analysis performed using Statistica v. 10.0 (StatSoft Polska, Kraków, Poland). They were recorded as means ± standard deviation (SD) and analyzed using Microsoft Excel 2007 software (Microsoft Corp., Redmond, WA, USA). Analysis of variance was performed with ANOVA procedures. Significant differences (*p* ≤ 0.05) between mean values were determined by Duncan’s multiple-range test. The Principal Component Analysis (PCA) was performed according to standard procedures using the Statistica program.

## 4. Conclusions

The conducted research showed that the variety of the elderberry had a greater impact on the basic chemical composition and the content of bioactive compounds than the place of harvesting (cultivation). This suggests the possibility of cultivating this species in urban conditions in order to obtain valuable fruit. In most cases, wild elderberry fruits had a higher content of bioactive compounds, which confirms the suitability of wild fruits for consumption. A few more of these compounds found in elderberries were collected in the Landscape Park than in the city center, which proves that this species copes well in the difficult conditions of the city’s microclimate. The antioxidant activity of the tested fruit variants indicated that in the wild cultivars, despite the high content of sugars, acids and carotenoids, it was low. Only the content of triterpenes correlated with antioxidant properties. The reason for this may be the lower polyphenol content of these fruits. They will be the subject of further research.

## Figures and Tables

**Figure 1 molecules-27-05050-f001:**
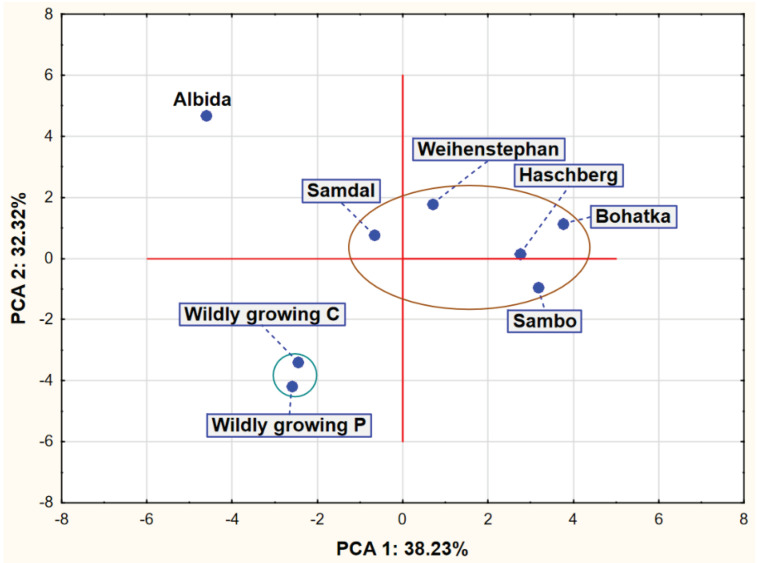
Plot of factor coordinates of cases in the PCA model.

**Figure 2 molecules-27-05050-f002:**
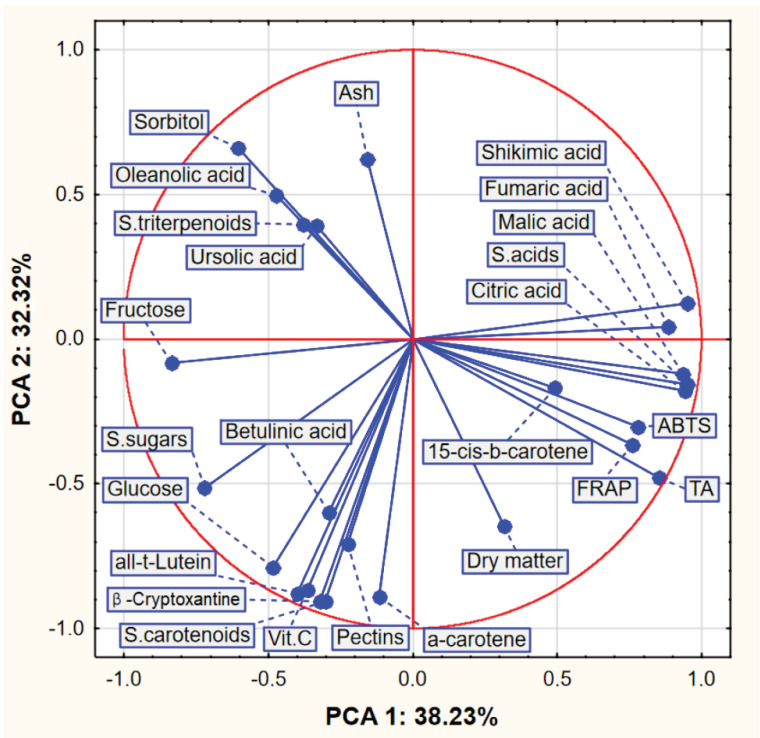
Plot of factor coordinates of variables in the PCA model.

**Table 1 molecules-27-05050-t001:** The content of dry matter, ash, acidity, vitamin C, pectins and antioxidant activity in the elderberry fruit.

Variety	Dry Matter	Ash	Titratable Acidity	Pectins	Vitamin C	FRAP	ABTS
	%	%	g/100 g FW *	%	mg/100 g FW	µMol/100 g FW	µMol/100 g FW
Albida	17.71 ± 1.98 f	1.25 ± 0.23 a	0.52 ± 0.08 d	0.83 ± 0.09 d,e	13.86 ± 2.14 c	1373.96 ± 48.09 e	707.26 ± 5.40 e
Bohatka	18.88 ± 2.47 e	0.86 ± 0.09 c	0.96 ± 0.09 a	0.88 ± 0.08 d	14.42 ± 1.68 c	4396.74 ± 201.9 c	2097.12 ± 15.63 c
Haschberg	22.32 ± 3.78 a	1.28 ± 0.16 a	0.93 ± 0.10 a	1.00 ± 0.09 b	15.70 ± 2.98 c	6426.89 ± 108.2 b	2849.66 ± 26.14 b
Sambo	20.72 ± 1.76 c	0.88 ± 0.09 c	0.94 ± 0.09 a	0.94 ± 0.09 c	23.28 ± 3.76 b	7276.70 ± 75.79 a	3359.04 ± 37.41 a
Samdal	20.04 ± 1.95 d	1.02 ± 0.09 b	0.82 ± 0.09 c	0.34 ± 0.08 f	10.19 ± 1.72 d	1819.41 ± 89.67 e	746.32 ± 3.25 e
Weihenstephan	21.14 ± 2,98 b	1.29 ± 0.09 a	0.87 ± 0.09 b	0.86 ± 0.09 d	8.56 ± 1.04 e	3669.43 ± 100.0 d	1684.81 ± 17.17 d
Wildly growing P	21.16 ± 4.25 b	0.93 ± 0.08 c	0.84 ± 0.09 c	1.54 ± 1.12 a	44.26 ± 5.98 a	3731.10 ± 99.95 d	1529.09 ± 11.92 d
Wildly growing C	21.13 ± 2.58 b	0.89 ± 0.07 c	0.82 ± 0.09 c	1.34 ± 1.06 a	39.67 ± 2.66 a	3685.19 ± 67.93 d	1678.04 ± 13.79 d

* Expressed as malic acid; ABTS, 2,2′-azino-bis-3-ethylbenzothiazoline-6-sulphonic acid; FRAP, ferric reducing antioxidant potential; mean values with different letters (a–f) within the same row were statistically different (*p* = 0.05). Values expressed as mean ± standard deviation; Wild-growing P—wild-growing in the Landscape Park, Wild-growing C—wild-growing P harvested in the city center.

**Table 2 molecules-27-05050-t002:** The content of sugars and organic acids in elderberries.

Variety	Fructose	Glucose	Sorbitol	Sum Sugars	Citric Acid	Malic Acid	Shikimic Acid	Fumaric Acid	Sum Organic Acid
	g/100 g FW				mg/100 g FW				
Albida	2.84 ± 0.16 a	1.31 ± 0.78 c	0.40 ± 0.09 a	4.55 ± 0.47 b	265.65 ± 19.56 e	99.56 ± 8.76 d	1.67 ± 0.12 c	0.87 ± 0.06 c,d	367.75 ± 32.85 c
Bohatka	0.99 ± 0.07 f	0.11 ± 0.09 e	0.00	1.11 ± 0.12 d	502.72 ± 46.32 a	141.87 ± 13.67 a	10.01 ± 0.97 a	4.34 ± 0.32 a	658.94 ± 62.84 a
Haschberg	1.93 ± 0.19 d	1.85 ± 0.08 b	0.00	3.79 ± 0.29 b	437.48 ± 29.63 b	139.85 ± 13.34 a	10.25 ± 1.09 a	3.72 ± 0.29 b	591.30 ± 45.32 a
Sambo	1.71 ± 0.12 c	2.02 ± 0.07 b	0.00	3.73 ± 0.26 b	466.23 ± 34.97 b	138.77 ± 13.21 a	9.45 ± 0.93 a	4.01 ± 0.30 a	618.46 ± 54.76 a
Samdal	1.92 ± 0.14 d	1.88 ± 0.09 b	0.00	3.79 ± 0.31 b	324.98 ± 30.74 d	105.12 ± 9.99 d	5.32 ± 0.51 b	1.06 ± 0.09 c	436.48 ± 38.65 c
Weihenstephan	1.23 ± 0.16 e	1.09 ± 0.09 d	0.00	2.32 ± 0.19 c	392.11 ± 28.81 c	122.18 ± 12.09 b	4.36 ± 0.39 b	1.02 ± 0.09 c	519.67 ± 49.12 b
Wildly growing P	2.42 ± 0.36 b	3.45 ± 0.19 a	0.00	5.87 ± 0.49 a	365.36 ± 29.63 d	113.63 ± 11.07 c	1.99 ± 0.13 c	0.95 ± 0.05 c	481.93 ± 39.67 c
Wildly growing C	2.25 ± 0.12 b	3.01 ± 0.08 a	0.00	5.26 ± 0.35 a	336.36 ± 26.23 d	111.99 ± 10.12 c	1.77 ± 0.09 c	0.90 ± 0.08 c,d	451.02 ± 35.02 c

Mean values with different letters (a–f) within the same row were statistically different (*p* = 0.05). Values expressed as mean ± standard deviation; Wild-growing P—wild-growing in the Landscape Park, Wild-growing C—wild-growing P harvested in the city center.

**Table 3 molecules-27-05050-t003:** Triterpenoid composition of different varieties of elderberry (µg/g of FW).

Variety	Betulinic Acid	Oleanolic Acid	Ursolic Acid	Sum
	µg/g FW			
Albida	49.27 ± 6.04 c	433.72 ± 52.71 a	1091.14 ± 110.6 a	1574.13 ± 123.2 a
Bohatka	44.15 ± 3.43 d	305.38 ± 32.96 b	925.71 ± 100.1 b	1275.24 ± 112.6 b
Haschberg	54.35 ± 4.27 b	172.62 ± 19.33 d	601.69 ± 59.94 d	828.66 ± 40.68 d
Sambo	50.14 ± 3.99 c	192.10 ± 20.73 d	579.28 ± 52.52 d	821.52 ± 76.34 d
Samdal	29.97 ± 2.51 e	137.61 ± 12.97 e	370.70 ± 32.87 e	538.28 ± 51.42 e
Weihenstephan	56.44 ± 6.01 b	237.93 ± 21.66 c	765.55 ± 61.63 c	1059.92 ± 100.9 c
Wildly growing P	68.65 ± 5.44 a	246.32 ± 19.83 c	734.64 ± 74.92 c	1049.61 ± 99.61 c
Wildly growing C	66.45 ± 5.27 a	239.49 ± 18.36 c	753.82 ± 24.88 c	1059.76 ± 102.6 c

Mean values with different letters (a–e) within same row were statistically different (*p* = 0.05). Values expressed as mean ± standard deviation; Wild-growing P—wild-growing in the Landscape Park, Wild-growing C—wild-growing P harvested in the city center.

**Table 4 molecules-27-05050-t004:** The content of carotenoids in the tested variants of elderberry.

Variety	All-Trans-Lutein	β-Cryptoxantine	α-Carotene	15-Cis-β-Carotene	Sum
	mg/kg FW				
Albida	4.67 ± 0.38 b,c	47.36 ± 4.44 f	0.00	0.00	52.03 ± 4.98 e
Bohatka	10.55 ± 1.05 b	280.66 ± 26.51 c,d	8.09 ± 0.87 b	29.55 ± 2.42 a	328.38 ± 51.9 c
Haschberg	7.61 ± 0.79 b	171.80 ± 15.51 d	1.92 ± 0.09 e	2.44 ± 0.12 d	183.77 ± 16.9 d
Sambo	9.56 ± 1.01 b	303.91 ± 29.76 c	6.29 ± 0.56 c	8.65 ± 0.82 b	328.42 ± 31.87 c
Samdal	14.84 ± 1.34 b	414.19 ± 39.72 b	4.43 ± 0.33 d	5.54 ± 0.53 c	439.00 ± 42.7 b
Weihenstephan	5.42 ± 0.46 b	127.63 ± 12.07 d,e	3.18 ± 0.29 d	1.55 ± 0.09 d	137.77 ± 12.8 d
Wildly growing P	31.50 ± 2.98 a	841.61 ± 7.45 a	14.08 ± 1.03 a	8.41 ± 0.74 b	895.60 ± 76.3 a
Wildly growing C	28.88 ± 2.53 a	789.58 ± 6.89 a	12.48 ± 1.19 a	8.58 ± 0.98 b	839.51 ± 79.9 a

Mean values with different letters (a–f) within the same row were statistically different (*p* = 0.05). Values expressed as mean ± standard deviation; Wild-growing P—wild-growing in the Landscape Park, Wild-growing C—wild-growing P harvested in the city center.

## Data Availability

The data is archived on the server of the University of Life Sciences in Wrocław.

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
