# Peer review of "Comparison of the Chemical Composition of Selected Varieties of Elderberry with Wild Growing Elderberry"

_molecules, 2022, doi:10.3390/molecules27165050_

Round 1

Reviewer 1 Report

General comments:

1. In general, this work describes and presents interesting results. The argumentation is good, but, in my opinion, some aspects can be improved:

A. More recent literature on this topic is welcome;

B. The analytical chemistry involved can be better described (in more detail). I'm talking about HPLC specificities... figures of merit, analytical curve., LOD, etc. Chromatograms are also welcome. More sampling details are also needed.

C. I suggest a multivariate approach to this work. The data are well presented and very interesting. A PCA and perhaps an HCA can help the authors in their argumentations and show possibly similarities (or differences) between the varieties studied.

Author Response

alla comments are use. We add data in fig. 1 and 2. 

Reviewer 2 Report

It is a relatively interesting manuscript that can be published after the following modifications:

1. Better describe the studied varieties of Sambucus nigra. Plant age, soil-climatic conditions, harvest time, etc. It is also not entirely clear how the samples were taken, from how many plants, how many replications there were, etc. How the samples were subsequently handled until the analyses...

2. The content of biologically active substances in fruits can be influenced not only by the cultivar, but also by the course of climatic conditions, nutrient content, and harvest time... This phenomene needs to be properly discussed.

3. If the samples of all studied cultivars were not taken at least in 2-3 seasons in order to reduce interannual variability and if the samples were not large enough, no clear conclusions can be drawn from the results. Accordingly, the conclusion should be formulated. Methodologically, the manuscript (if the authors do not refute this premise) would more closely correspond to the Short Communication.

Author Response

All comments are used

Data comes from years 2019 - 2021, but we prezesent it as a average of three years.

Plants were grown in a natural, bushy form. All the bushes were fully fruiting. Their age is 5-10 years. The plants were grown on sandy, light soils, surrounded by other fruit crops. 

All plants grew naturally, accompanying other fruit crops. No additional fertilization or plant protection. The fruits were harvested in the second decade of September, after all the varieties were fully ripe. Maturity was assessed on the basis of easy fruit detachment from the stalk. Twelve plants were sampled for each cultivar, and the tests were carried out with 4 replications per combination. After harvesting, the fruit was frozen, and then the homogenization of the plant material was commenced for the purpose of making tests. Whole fruits were analyzed, without petals.

The course of the weather in the research years does not differ significantly from the long-term averages. The average temperature in July in all surveyed sites was between 18 and 19,5 C. There were no periods of drought or excessive rainfall.

The presented results are average data from three years of sampling (2019-2021). The same analyzes were performed each year and then averaged in order to present the results of studies representative for three fruiting seasons.

Round 2

Reviewer 1 Report

Yes, the manuscript has been sufficiently improved to warrant publication in Molecules. 

Reviewer 2 Report

The authors corrected the manuscript according to the reviewer's comments, I have no further comments.